# An analytical delay model for multi-class and lane-free traffic condition

**Vinaya S. Mattungal** [ID]◉, **Lelitha Devi Vanajakshi** [ID]◉*

Department of Civil Engineering, Indian Institute of Technology Madras, Chennai, Tamil Nadu, India

◉ These authors contributed equally to this work.
* lelitha@iitm.ac.in

## Abstract

This study emphasises the criticality of delay as a performance metric for signalized intersections and the challenges associated with its estimation, particularly in the context of Multi-class and Lane-free (MCLF) traffic conditions. Traditional delay models are often inadequate for such conditions, necessitating the development of a tailored approach. A novel delay equation is proposed, integrating insights from queuing theory principles with consideration of multi-class of vehicles and lane-free movement. Key features include assumption of random arrival and departure pattern as well as distribution, incorporation of Passenger Car Equivalent (PCE) and virtual lane concepts to account for the diverse vehicle classes and lane-free movement prevalent in Indian traffic. The model's efficacy is demonstrated through comparison with conventional in practice delay models, showing its superior performance. This tailored approach enhances the accuracy of delay estimation and also highlights the importance of accounting for specific traffic characteristics in optimising signal design for intersections under MCLF traffic conditions.

## Introduction

Vehicle delay is the most important measure for characterising the performance of signalized intersections [1,2]. Its significance is evident in its widespread application across both design and evaluation process. It effectively measures the inconvenience imposed by traffic signals on road users and can even be translated into monetary values [3]. Additionally, this measure is employed to measure the exhaust emissions and fuel consumption.

The delay at a signalised intersection is defined as the additional travel time experienced by a vehicle due to the intersection control, compared to if it were uncontrolled. It encompasses the lost time due to deceleration, acceleration and stopping. However, delay is a parameter that is challenging to ascertain due to the unpredictable nature of the arrival and departure (service) processes of vehicles at the intersection. Hence, number of analytical delay models are proposed to calculate delay at the intersection.

However, majority of the existing delay models have been derived and calibrated for homogeneous and lane disciplined (HoLD) traffic conditions [4–7]. In many of the developing countries including India, traffic conditions are multi-class and lane-free (MCLF). Here, "multi-class" traffic refers to a mixture of different types of motorised and non-motorised vehicles using the same roadway and "lane-free" indicates that vehicles often do not adhere

**Data availability statement:** All relevant data are within the manuscript and its Supporting Information files.

**Funding:** Ministry of Road Transport and Highways (MoRTH). The funders had no role in study design, data collection and analysis, decision to publish, or preparation of the manuscript.

**Competing interests:** No authors have competing interests.

to lane discipline most of the time. This practice often leads to small vehicles attempting to move to the front of queues by exploiting available gaps in both lateral and longitudinal directions, disregarding lane discipline. Consequently, this behaviour results in numerous conflicts, ultimately reducing intersection capacity and causing increased delays. Since these are not considered in the existing delay models, they may not accurately estimate the delay under such conditions. Consequently, there is a need for an analytical delay model specifically tailored to accommodate the traffic dynamics observed in MCLF traffic conditions and is the aim of this study.

To achieve this goal, the paper presents a delay model derivation tailored to fit the MCLF traffic conditions. This model incorporates multi-channel queueing theory, Passenger Car Equivalent (PCE), virtual lane concept, and considers the arrival and service distribution of vehicles. The microsimulation evaluation of the developed model reveals its superior performance compared to conventional delay models. The remainder of the paper is structured as follows; in the subsequent section the existing literature on different delay models are discussed. The preliminary analysis and methodology of the proposed framework for the derivation of the delay equation is discussed next. This is followed by the validation of the developed model and the last section summarises the significant findings from the research.

## Literature review

Travel time and delay are the fundamental measures of level of service for any transportation system and numerous studies are conducted to estimate these variables [8–10]. Initial delay estimation methods relied on historic data and utilised queueing theory [11] and shockwave analysis [12]. Another common approach is by employing the arrival and departure counts of vehicles, cumulative count curves are generated to calculate the delay. The area between the arrival and departure curve yields the total travel time. The average delay per vehicle can be obtained from the area of the cumulative count curves by shifting the arrival curve by free flow travel time. However, this method requires precise count of arrival and departures of vehicles, which is difficult to obtain from field [5]. Analytical delay models can be grouped as deterministic and stochastic based on whether they take into account the random characteristics of traffic flow. The stochastic element of delays utilizes the distributions for traffic arrival and service times. Models that integrate both deterministic (referred to as uniform) and stochastic (random or overflow) aspects of traffic performance are particularly attractive in the realm of traffic signals, as they can be applied to a broad spectrum of traffic intensities and various types of signal controls, and are discussed below.

### Deterministic delay models

The deterministic queuing models can accurately estimate delay at signalized intersections, where green interval serve a greater number of vehicles than the number of arrivals per cycle [13]. The area between the cumulative arrivals and departures are considered as the deterministic component of delay [14] and is estimated according to the following assumptions: i) zero initial queue at the start of the green phase, ii) uniform arrival pattern at the arrival flow rate ($q$) throughout the cycle iii) uniform departure pattern at the saturation flow rate ($S$) during queue discharge, and at the arrival rate when the queue vanishes, and iv) arrivals do not exceed the signal capacity, defined as the product of approach saturation flow rate ($S$) and its effective green to cycle ratio ($g/C$). Newel [15] employed the continuum model, using $A(t)$ and $D(t)$ to represent cumulative arrivals and departures at any time $t$. The arrival curve is shifted by the free flow travel time to get the virtual arrival curve. The area between the virtual arrival and departure curve for a cycle gives the total delay per cycle. May and Keller [16] developed

a delay model with a deterministic assumption for over-saturated conditions. In reality, the randomness of traffic may cause fluctuations, rendering the deterministic assumption invalid.

## Steady-state stochastic models

The deterministic models rely on the assumption of uniform arrivals, while stochastic delay models aim to incorporate the variability associated with vehicle arrivals [17]. Beckmann et al. [18] proposed a stochastic model for vehicle delay with binomial arrivals and uniform departures. Newell [19] applied the Beckmann model to various arrival rates and demonstrated that binomial arrivals, which ensure a minimum spacing between vehicles, prevent infinite delays as $q/s$ approaches unity. Although this model provided higher delay estimates than those based on deterministic arrivals, it still underestimated the observed delay values [20]. Later, vehicle arrivals at intersections were modelled as a Poisson process [21]. The model incorporated a random delay term in addition to a uniform delay term to calculate the total delay per vehicle. Webster [22] modified the above M/D/1 model by adding empirical terms to Kendall's model. It is comprised of three components. The first component estimates delay assuming a uniform arrival rate, while the second term accommodates the randomness in arrivals, known as the "random delay", which is derived for a Poisson arrival process and services at a constant rate equivalent to signal capacity. The third component, calibrated through simulation experiments, is an adjustment factor and typically in the range of 5-15 percent of the first two components.

Subsequent to Webster's work, several other stochastic models emerged [15,23–25]. Miller [23] developed a delay model based on a general arrival process and a deterministic service process, introducing the variance-to-mean ratio ($I$) to account for the random nature of arrivals. Miller's model gave better results compared to Webster's model when the value of $I$ is greater than one [15,20]. Later, Hutchinson [26] suggested modifications to the random delay term of Webster's model by including the $I$ ratio, resulting in more accurate delay estimates than Webster's model. In comparison to binomial and uniform arrival types, the Poisson assumption yields better estimates when the degree of saturation is less [20]. However, the limitation of the Poisson assumption is that it permits for small time headway between arrivals, which can lead to infinite queue lengths and delays when the degree of saturation approaches unity [19]. These models share common basic assumptions. They all assume that the departure headways follow a known distribution with a constant mean. Furthermore, it is assumed that the system remains under-saturated throughout the analysis period, despite temporary oversaturation due to the variability in arrivals. Finally, it is presumed that the system has been operational for a duration sufficient to stabilise into a steady state.

However, in reality, the assumptions of steady-state models are compromised when the flow rate fluctuates due to randomness in arrivals, as stochastic equilibrium cannot be reached. Thus, there is a necessity for developing delay models taking into account these unique features, and is addressed in this study.

## Time-dependent delay models

Several attempts have been made to address the limiting assumption of steady-state conditions. One of the best approaches is to approximate time dependent arrival profile by some mathematical function and calculate corresponding delay. This method connects the steady-state stochastic delay model with the deterministic over-saturation model. Catling [27] and Akcelik [28] introduced the coordinate transformation technique to develop such a model. This technique connects the steady-state stochastic delay models to a deterministic over-saturation delay developed by May and Keller [16]. However, the deterministic oversaturation

delay model overlooked initial queue delay, resulting in Catling and Akcelik's model also failing to capture this aspect. Kimber and Hollis [29] introduced a time-dependent delay model based on the assumption of random arrivals and general departures with one server system. The capacity guide delay models currently utilised in the United States [30], Australia [31], and Canada [32] are examples of time-dependent delay models. Akcelik [33] and Burrow [34] proposed generalised delay models from which the above models could be obtained by substituting suitable parameter values. Akcelik [33] demonstrated that the HCM 1985 [30] model overestimated the delay in over-saturated scenarios due to the $X^2$ term in the overflow delay component of the model. Akgungor and Bullen [10] recommended replacing $X^2$ with $X^n$ and also derived a non-linear function to calculate $n$. Fambro and Rouphail [9] suggested incorporating analysis period $T$ in the HCM delay model and also proposed a third term ($d_3$) to accommodate initial queue delays. The suggestions were adopted in the HCM 2000 [35] model.

However, all of the above studies are for HoLD traffic conditions, with obvious limitations for using under MCLF traffic conditions. Specific characteristics of MCLF traffic with a wide variety of vehicle classes with lane free movement may need to be incorporated in the model for it to perform well. One of the main characteristics is the smaller vehicles squeezing through the space between the larger ones, flouting the first-in, first-out (FIFO) queueing rule. A lot of parallel movements occur both within and across lanes as a result of lane-free movement [36]. Unlike conventional car following, gap filling emerges as the predominant behaviour in such scenarios [37]. The movement of vehicles can be affected by factors such as geometric design, signal control settings, driver behaviour and traffic conditions [38]. These behaviours are different from those seen in HoLD environment and need to be incorporated in the model for it to perform well under such traffic conditions. Researchers have attempted to modify the conventional delay models to better accommodate MCLF traffic conditions are discussed next.

## Delay models for MCLF traffic conditions

All of the above discussed models are derived and calibrated for HoLD traffic condition and their applicability under MCLF traffic conditions are limited. Hoque and Imran [39] proposed modifications to Webster's model by modifying the third component using multi-linear regression. Preethi et al. [7] introduced a semi-empirical adjustment term to the Webster's model, which was derived from field observations utilising Artificial Neural Networks (ANN). However, such attempts to adapt Webster's delay model for MCLF traffic conditions by introducing calibration constants may not be sufficient, as the fundamental assumptions in the Webster's method remain the same.

In MCLF traffic condition, two-wheelers often navigate through gaps between larger vehicles to leave intersections earlier. Thus, a higher percentage of two-wheelers can result in a random service distribution. An M/M/1 or M/M/n model with random delay term would be more suitable for such scenarios. Medhi [40] derived the delay expression for an M/M/n queueing system with FIFO queue discipline. Later, Verma et al. [6] proposed a modified Webster's delay model by comparing the delay values with HCM field estimation method. They assumed an M/M/n queueing system with Probably-First-In-Probably-First-Out (PFIPFO) queue discipline. Despite attempting to modify the service process, the authors did not measure the effect of PFIPFO approach as an alternative to traditional FIFO assumption, as the random delay term used was same as the one for FIFO [40]. Moreover, the delay model is cumbersome to solve analytically to get a closed form expression. Mukhopadhyay et al. [41] extended Webster's mean delay formula to account for MCLF traffic conditions by

incorporating a Markovian service process. However, their analysis assumes a single-server scenario, which does not accurately reflect the characteristics of MCLF traffic conditions. Khadhir et al. [42] proposed a theoretical delay model based on queuing theory incorporating the characteristic features of MCLF traffic conditions such as lane free movement, flouting the FIFO rule and different composition of vehicle types. The second term of the delay model, which is the random delay, which is the average delay expression of M/D/n queuing model. Thus, it assumes deterministic vehicle service distribution, which may not be true under varying traffic conditions, especially under MCLF condition [17].

From the literature review it is apparent that, the majority of the theoretical delay model are tailored for HoLD traffic conditions and their applicability in MCLF traffic condition is limited. Consequently, straight forward application of these conventional delay models in MCLF traffic condition can cause inaccurate delay estimate. A few studies have attempted to recalibrate the conventional delay models for MCLF traffic conditions. These models may not be sufficient as the fundamental assumptions being the same. The delay models using the M/D/n queueing system may not accurately estimate delay under MCLF conditions, especially for high degree of saturation or with significant two wheelers presence. Studies that developed using M/M/n queueing theory was not validated sufficiently and did not explicitly incorporate the unique features of MCLF traffic condition with random delay term same as that of FIFO. Moreover, the delay model is cumbersome to solve analytically to get a closed form expression.

To overcome this gap, this research develops theoretical delay model for MCLF traffic condition by incorporating the traffic characteristics such as lane free movement and multi classes of vehicles. The queueing system, queueing process, number of servers, arrival and departure pattern as well as their respective distribution also were chosen based on the MCLF traffic condition observed from the field. The resulting delay model is a closed form expression that can be further solved analytically.

## Study site, data, and preliminary analysis

Two intersections were identified for data collection, representing MCLF traffic conditions from two different cities in India. The first study area chosen is Mhalgi nagar square intersection in Nagpur, a four-legged intersection with 6 lane major road and 4 lane minor road crossing at level grade. The location of the intersection and snap shot of the video footage are shown in Figs 1(a) and 1(b) respectively. The intersection operates under fixed time traffic signal. Video data collected during the development of Indo-HCM [43] was utilised for the current study. The data was collected during the morning peak hour of 10:00 AM to 11:00 AM. From the videos, classified vehicle counts and the distribution of turning movements were manually extracted. The vehicle composition was found to be 86% two-wheelers (motor bike), 9% cars, 4% three-wheelers (auto-rickshaws) and 1% heavy vehicles (bus and truck).

Traffic signal timings were also noted down, specifically the duration of green lights for each phase. The average signal cycle lasted 125 seconds, with green light durations of 32, 25, 35, and 25 seconds allocated to approaches 1, 2, 3, and 4, respectively. The saturation flow of the study approach was extracted manually from the video footage by counting the number of vehicles crossing the stop line during the queue dissipation period (5 second interval), after the initial lost time. These counts were then converted to equivalent values using the PCE values developed by Khadhir et al. [42] and then to Through Car Units (TCU). The resulting saturation flow was found to be 5200 TCU/hour of green/lane.

The second study site selected is the Kaiveli intersection in Chennai, India. It is a three-legged intersection operating under fixed time signal control. The major road is 9.9 meters

A

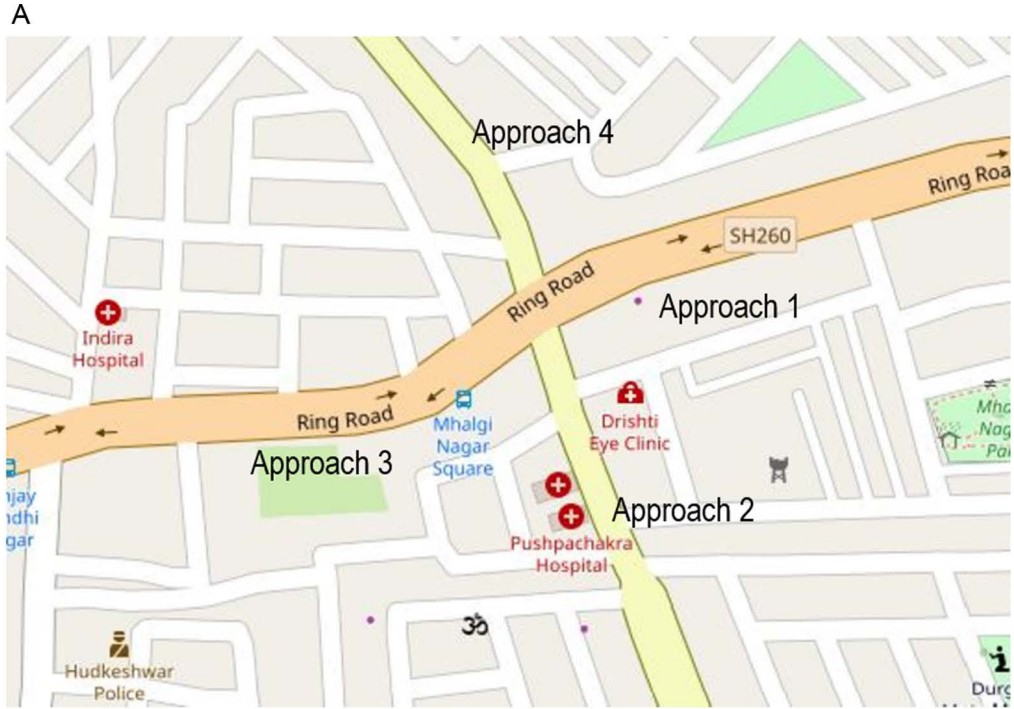

B

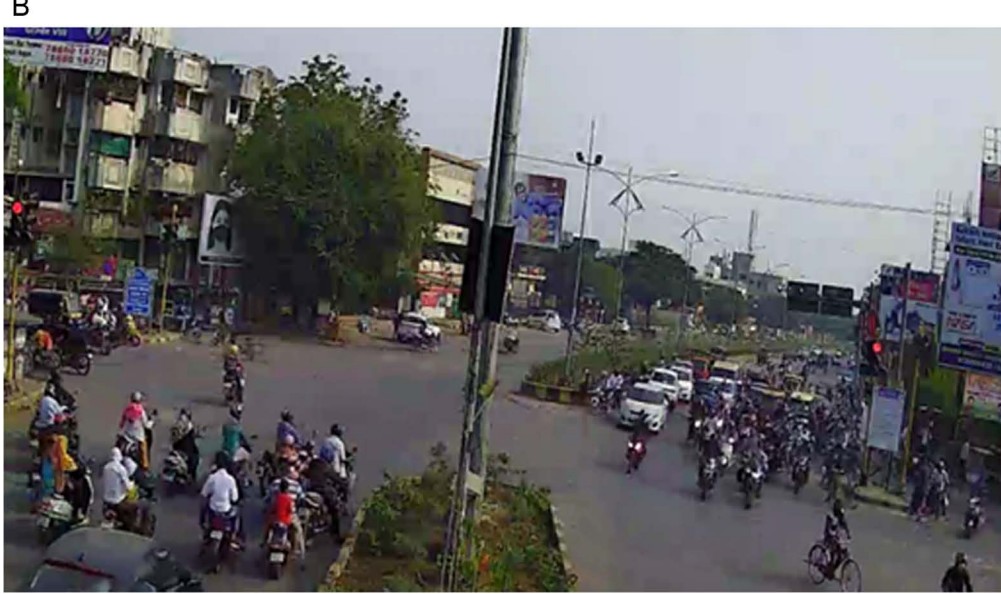

**Fig 1.** (a). Location of study site (Mhalgi nagar square intersection, Nagpur). (b). Snapshot of video footage of Mhalgi nagar intersection.

wide in a single direction, with a median and shoulders. Drone video data collection was conducted in the study site during the noon peak period of 12:15 PM to 1:15 PM and the analysis was carried out for North bound through movement traffic. The location of the intersection and snap shot of data collection are shown in the Figs 2(a) and 2(b) respectively.

The traffic flow and signal timings were extracted from the video. All the signal cycles were under-saturated without any residual queue. The cycle length was 125 seconds and green

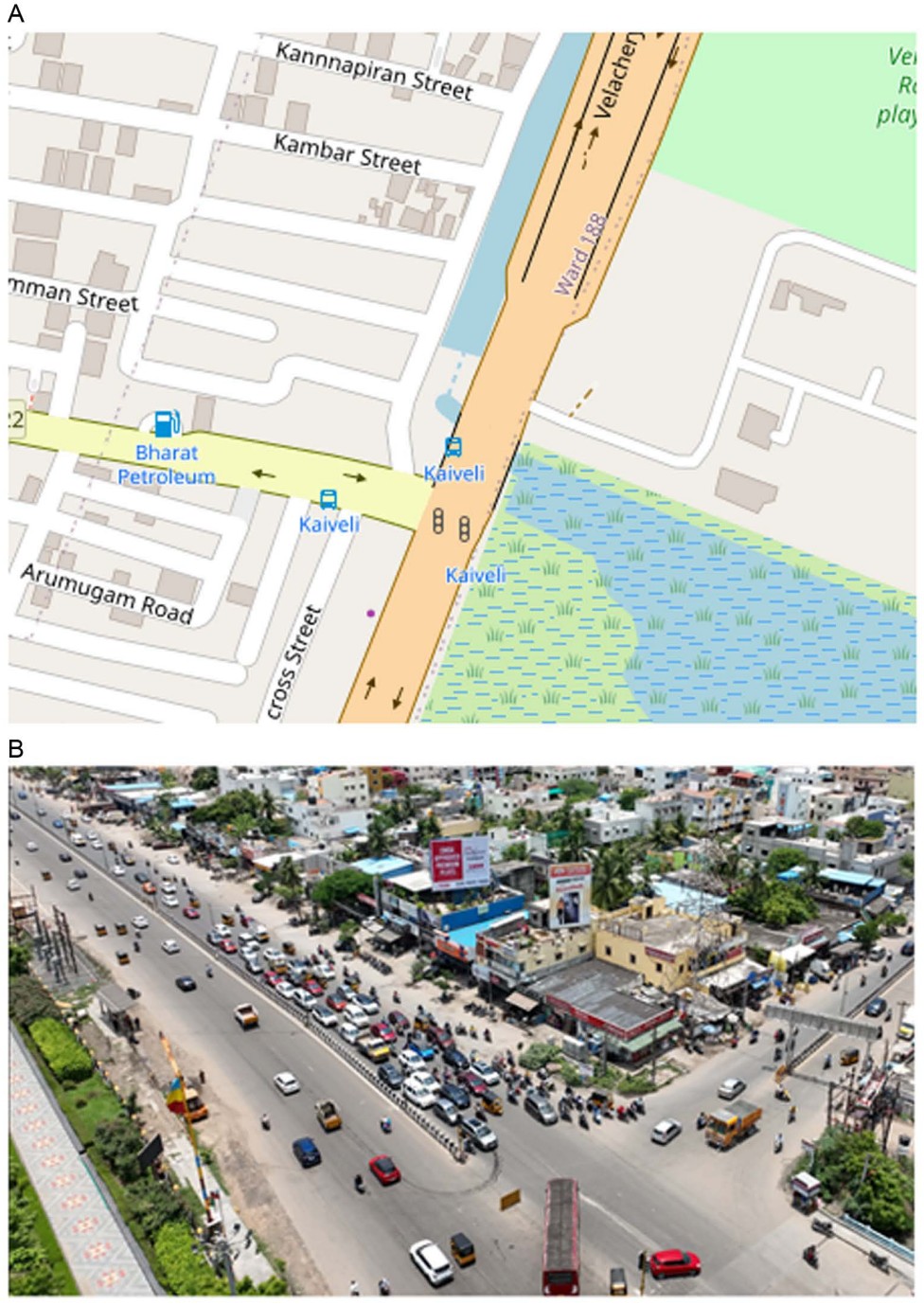

**Fig 2.** (a). Location of study site (Kaiveli intersection, Chennai). (b). Snapshot of video footage of Kaiveli intersection.

duration of north bound approach was 65 seconds. The classified vehicle counts revealed an average vehicle composition of 60% two-wheelers (motor bike), 29% cars, 7% three-wheelers (auto-rickshaws) and 4% heavy vehicles (bus and truck). Field data indicated a saturation flow of 3900 PCE/hour of green/lane.

The traffic demand extracted from the video footages for sample approaches are presented in Tables 1 and 2 for Mhalgi nagar and Kaiveli, respectively. These data are utilised for the performance comparison of conventional delay models and preliminary analysis, discussed in the following sections.

Table 1. Traffic demand for approach 1 in morning peak hour (10:00 AM to 11:00 AM) at Mhalgi nagar, Nagpur.

| Cycle Number | Approach demand (veh/hour) |
| --- | --- |
| 1 | 3110 |
| 2 | 3370 |
| 3 | 3456 |
| 4 | 3542 |
| 5 | 3485 |
| 6 | 3542 |
| 7 | 3946 |
| 8 | 3341 |
| 9 | 3888 |
| 10 | 4493 |
| 11 | 3802 |
| 12 | 4147 |
| 13 | 4262 |
| 14 | 3686 |
| 15 | 2621 |
| 16 | 3283 |
| 17 | 3226 |
| 18 | 2275 |
| 19 | 3024 |
| 20 | 3197 |
| 21 | 2851 |
| 22 | 2534 |
| 23 | 2592 |
| 24 | 3312 |
| 25 | 2765 |
| 26 | 2707 |

Table 2. Traffic demand for north bound approach in afternoon peak hour (12:15 PM to 1:15 PM) at Kaiveli, Chennai.

| Cycle Number | Approach demand (veh/hour) |
| --- | --- |
| 1 | 6270 |
| 2 | 5428 |
| 3 | 5317 |
| 4 | 5566 |
| 5 | 6480 |
| 6 | 6397 |
| 7 | 4957 |
| 8 | 5511 |
| 9 | 5650 |
| 10 | 4929 |

For the validation, the average vehicular delay experienced during this time was also extracted. To achieve this, the time at which individual vehicles came to a stop and the time they started moving for each signal cycle were meticulously extracted for one approach. The acceleration and deceleration delays were calculated by considering their respective rates [44]. These were added to the stopped delay and then were used for validation.

Next, some of the commonly used delay models, namely Webster's delay model [22], Akcelik delay model [31], HCM model [35], and Indo-HCM delay model [43] have been chosen and their performance was evaluated under MCLF traffic condition. The respective equations of these delay models are shown in equations (1) to (4).

$$d = \frac{C(1-\lambda)^2}{2(1-\lambda X)} + \frac{X^2}{2q(1-X)} - 0.65\left(\frac{C}{q^2}\right)^{\frac{1}{3}} X^{(2+5\lambda)}. \tag{1}$$

$$d = \frac{C(1-\lambda)^2}{2(1-\lambda X)} + 900T\left[(X-1) + \sqrt{(X-1)^2 + 12\left(\frac{X-X_0}{cT}\right)}\right]. \tag{2}$$

$$d = \frac{0.5C\left(1-\frac{g}{C}\right)^2}{1-\left[\min(1,X)\frac{g}{C}\right]}(PF) + 900T\left[(X-1) + \sqrt{(X-1)^2 + \frac{8kIX}{cT}}\right] + \frac{1800Q_b(1+u)t}{cT}. \tag{3}$$

$$d = \frac{0.5C\left(1-\frac{g}{C}\right)^2}{1-\left[\min(1,X)\frac{g}{C}\right]}(0.9) + 900T\left[(X-1) + \sqrt{(X-1)^2 + \frac{8kIX}{cT}}\right] + \frac{1800Q_b(1+u)t}{cT}. \tag{4}$$

A list of notations used in these equations, and eventually in this paper also, along with their description is provided in Table 3. The choice of these models stems from their widespread use among practitioners and researchers globally, with the fourth model specifically tailored for Indian MCLF traffic conditions.

In the above models, the first term denotes the uniform delay component. The Webster's delay model features the second term as the random delay factor, while the other three models incorporate an overflow delay component in the second term. This overflow delay encompasses both continuous oversaturation and temporary oversaturation resulting from random variations in arrivals. Additionally, HCM and Indo-HCM models, comprised of a third term signifies the initial queue delay. The HCM and Indo-HCM model exhibits considerable similarity, with the exception being the utilisation of a PF value of 0.9 specifically in the Indo-HCM model.

In order to evaluate the performance of these selected delay models in MCLF traffic condition, the delay estimates from these conventional delay models should be compared with the delay observed in the field. However, collecting data on intersection delay for each cycle directly from the field is challenging. Hence, field traffic conditions at the two study sites were simulated to generate equivalent delay values. For this, the microscopic simulation software, VISSIM version 11 was used. Indian traffic, with MCLF traffic conditions, requires adjustments in VISSIM parameters to accurately represent its characteristics. VISSIM features such as accommodating left-side driving, various vehicle types, flexible lane width within the same road, adjustable lateral distance between vehicles, allowance for overtaking within lanes, and staggered queueing at intersections enhance its suitability for modelling in MCLF Indian

**Table 3. List of notations.**

| Notation | Description |
|---|---|
| **Signal control related variables** | |
| $C$ | Cycle length (sec) |
| $R$ | Effective red time for a phase (sec) |
| $g$ | Effective green time for a phase (sec) |
| $\lambda$ | Proportion of the effective green to cycle length (g/C) |
| $q$ | Traffic demand (vehicles/hour or PCE/hour) |
| $s$ | Saturation flow rate (vehicles/hour or PCE/hour) |
| $c$ | Capacity rate (vehicles/hour or PCE/hour), c = sg/C |
| **Theoretical delay model variables** | |
| $d$ | Average delay on a single approach to an intersection (sec/vehicles or sec/PCE) |
| $d_1$ | Uniform delay (sec/vehicles or sec/PCE) |
| $d_2$ | Random delay (sec/vehicles or sec/PCE) |
| $d_3$ | Initial queue delay (sec/vehicles or sec/PCE) |
| $PF$ | Progression Adjustment Factor |
| $I$ | Ratio of variance to mean count of vehicles/Upstream filtering/metering adjustment factor |
| $T$ | Analysis period duration (hour) |
| $X_0$ | Degree of saturation below which the overflow delay is negligible in capacity guide models |
| $Q_b$ | Initial queue at the start of analysis period T (vehicles) |
| $u$ | Delay parameter |
| $k$ | Incremental delay factor dependent on controller settings |
| $X$ | Degree of saturation, ratio of actual flow to capacity, $X = q/c$ Utilisation factor in the context of queueing theory. For a multi-server system, $X = q/n\mu = q/c$ |
| **Queueing theory related variables** | |
| $W_q$ | Average waiting time in the queue |
| $n$ | Number of service channels or number of virtual lanes |
| $\mu$ | Discharge rate (vehicles/hour or PCE/hour) |
| $N$ | Number of entities in the system |
| $P_o$ | Probability of zero entities in the system |
| $P_N$ | Probability of N entities in the system |
| $p$ | Average processing time (s) |

traffic [45,46]. Optimal values of these parameters are taken from previous studies that consider MCLF traffic conditions [4,45,47,48]

To validate the calibrated VISSIM network for the selected intersections, the travel time between two points along the data collection stretch was utilised. The travel time over the duration of 10 cycles was noted. The corresponding travel time was recorded using the "Vehicle travel time measurement" tool within VISSIM. These values were compared and the error was quantified using Mean Absolute Percentage Error (MAPE), which was observed to be 2.01% for one location and 6% for the second location. These results indicate that VISSIM is capable of accurately reproducing MCLF traffic conditions observed in the field.

After calibrating and validating the VISSIM network, five simulation runs were conducted to compute the average control delay for each cycle. The output was generated for 1s interval and it was then aggregated over the cycle time to get the cycle wise results. The cycle wise results were weighted by the number of vehicles observed in each cycle to get the average approach control delay. Subsequently, these values were compared with the delay estimates from the chosen delay models. The cycle-by-cycle comparison of the delay results of Mhalgi nagar square intersection and Kaiveli intersection are depicted in Figs 3 and 4 respectively.

It is evident from the figures that none of the conventional models closely match the field delay estimates, making them inadequate for estimating delay under MCLF traffic conditions. The MAPE values relative to delay values of Mhalgi nagar intersection were 40%, 40%, 49%, and 45% for the Webster, Akcelik, HCM, and Indo-HCM models, respectively. The MAPE values for Kaiveli intersection were 15%, 15%, 42%, and 22% for the same models. These discrepancies highlight the need for a delay model that takes into account the specific characteristics of MCLF traffic conditions.

## Methodology

This study adopts queueing theory-based approach to derive an expression for delay in MCLF traffic conditions. A queueing system composes of a server, a stream of customers who requires service and a queue or line of customers waiting to be served [49]. Each of these elements considered in this study are discussed in detail in the following sections in the context of MCLF traffic condition.

### Arrival and departure rate

It is the rate at which vehicles arrive and depart the system expressed in vehicles/ hour. In the context of MCLF traffic, it should be expressed with consideration of multiclass of vehicles.

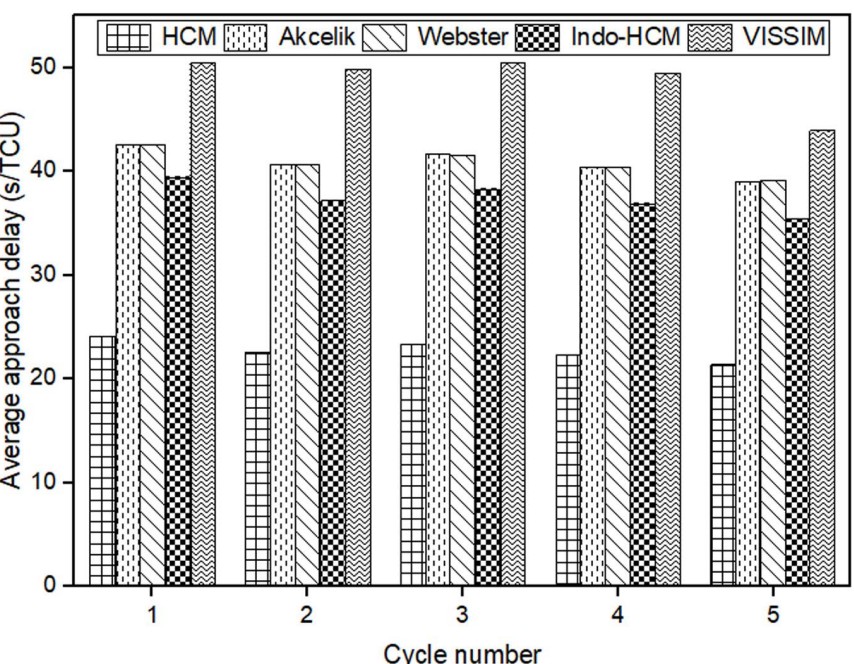

**Fig 3. Comparison delay estimates of conventional delay models with delay values from Mhalgi nagar VISSIM network.**

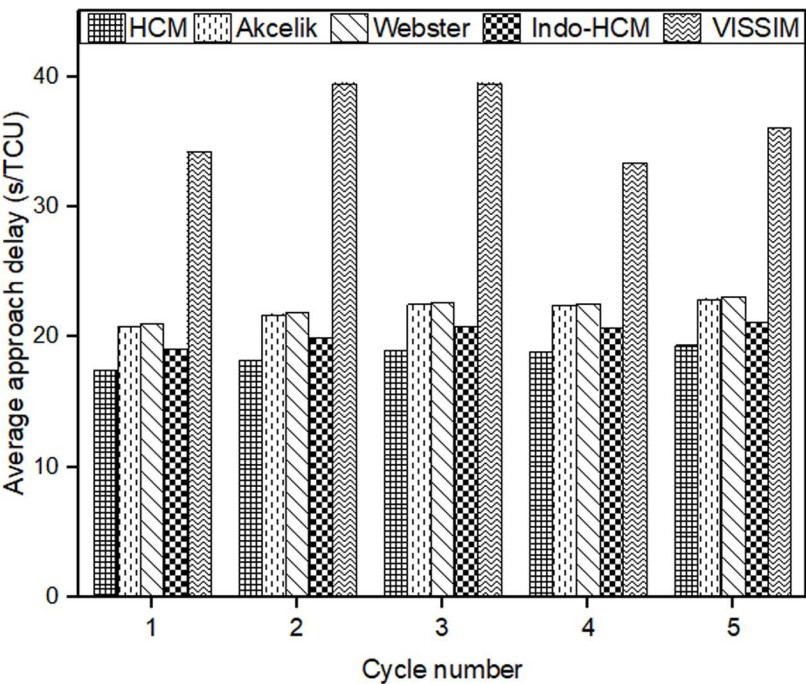

**Fig 4. Comparison delay estimates of conventional delay models with delay values from Kaiveli VISSIM network.**

There are different approaches for taking into account the multiple classes of vehicles and this study used the PCE concept. Most of the PCE estimation method in MCLF traffic condition is based on delay, headway, saturation flow, travel time and queue discharge rate [50,51]. As the saturation flow is a characteristic of the intersection, PCE estimation based on saturation flow [42] is adopted in this study. The PCE values derived based on the concept of minimisation of saturation flow between inter-cycle is used as given in Table 4.

## Arrival and departure pattern

At an intersection, vehicles can be queued and discharged in many ways, each affecting traffic flow differently. Some of the possible scenarios include:

**Single channel-single queue.** The vehicles approach the intersection in separate lanes and discharge from the same lane in the order in which they have arrived. Each lane has single queue of vehicles, and there is no cross over between the lanes. This scenario is illustrated in Fig 5.

**Single channel-single queue through virtual lanes.** The vehicles approach the intersection through number of virtual lanes which do not correspond to marked physical lanes. The vehicles discharge through the same virtual lanes, as illustrated in Fig 6. The

**Table 4. PCE values of vehicles.**

| Vehicle type | PCE values |
|---|---|
| Car | 1 |
| Two-Wheeler (Motor bike) | 0.78 |
| Three-wheeler (Auto rickshaw) | 1.92 |
| Heavy vehicles (Bus and Truck) | 3.42 |

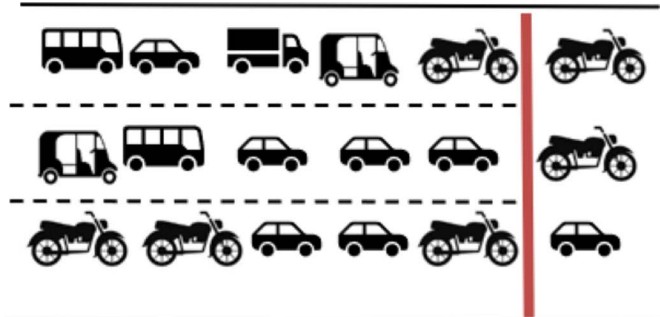

**Fig 5. Lane disciplined vehicle movement through physical lanes.**

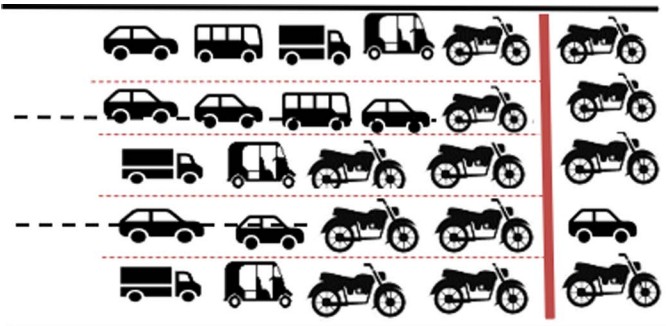

**Fig 6. Lane disciplined movement through virtual lanes.**

physical lane and virtual lanes markings are indicated by black and red dotted lines respectively.

**Multichannel- single queue.** Vehicles approach the intersection following lanes through clearly defined physical lanes. However, while queueing, they fill available space forming virtual lanes and discharge as depicted in Fig 7.

**Multichannel- multi queue.** Vehicles approach the intersection in a lane-free manner, which can be conceptualised as a single virtual lane for each approach. During the discharge process, this virtual lane splits into multiple virtual lanes to facilitate smooth traffic flow. This scenario, where the single virtual lane splits into six virtual lanes during discharge is shown in Fig 8.

Out of these, the best way to explain the queue formation under MCLF traffic conditions is the last scenario of a single virtual lane splitting into a number of parallel virtual lanes. This happens due to the multi class of vehicle types, all adhering to lane free movement. Smaller vehicles such as two-wheeler and three-wheeler navigate through the gap between the larger vehicles, often creating a virtual lane for their travel.

## Number of servers

The smaller vehicles often squeeze through the space between the larger ones, flouting the FIFO queueing rule. Hence, the number of lanes marked on the road will not be equal to the number of service channels. Virtual lane concept is utilised to find the number of servers occurring in the lane-free movement. The number of servers is based on the number of vehicles crossing the stop bar in parallel at any given moment. Since the composition of vehicles crossing the stop bar varies over time, this number fluctuates.

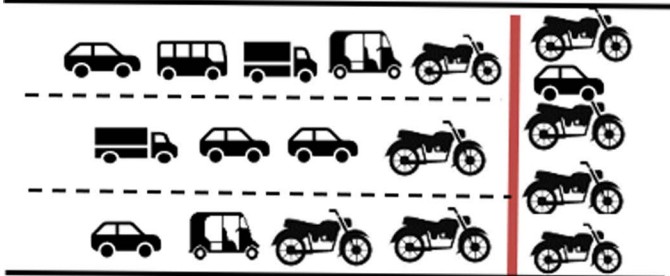

**Fig 7. Single queue splits in to multi channels.**

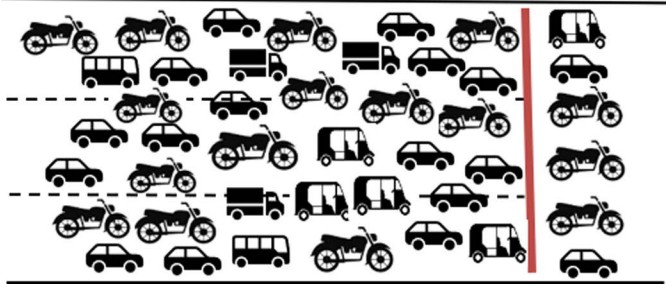

**Fig 8. Multichannel formation from multiple queues.**

To establish an average value for the number of servers, the average number of parallel movements is considered at three different times during the queue dissipation process: (a) the onset of queue dissipation (5 seconds after the start of the green signal), (ii) the midway through the queue (at 30% of the green time), and (iii) upon completion of queue dissipation [42]. It is assumed that the number of virtual lanes remains constant for a given road geometry, vehicle composition, traffic, and control conditions. Table 5 shows an example of the calculation of virtual lanes for the Kaiveli study site. The actual number of lanes was 3, even though the effective number of service channels increased to 7 due to parallel movements. Similarly, the number of virtual lanes was calculated at the Mhalgi nagar intersection, where 3 lanes were functioning as equivalent to 5 service channels.

## Queueing model

Queueing models are the mathematical representations used to analyze the behavior of queues. There are different types of queueing models, each characterized by specific features such as distribution of arrivals and departures, number of servers, queue discipline, system capacity and population size. These characteristics are discussed here.

**Statistical distribution of arrival and departure.** To develop an appropriate delay model for the MCLF traffic conditions, it is crucial to conduct a thorough analysis of the distribution patterns of arrivals and services. Since the arrivals are generally not uniform, the Poisson distribution is assumed in all the convention delay models mentioned above. In instances where arrivals are unscheduled, it is mathematically convenient to assume it as random, implying that they are equally likely to occur at any time [52]. In MCLF traffic condition also, the arrivals are typically random [6,42] However, these conventional models also assume uniform departure distribution, which may not always hold true in

**Table 5. Calculation of parallel servers for the Kaiveli study site.**

| Cases | Two-wheeler | Three-Wheeler | Car | Heavy vehicles | Total PCE | No. of parallel servers |
|---|---|---|---|---|---|---|
| Onset of queue | 5 | 1 | 0 | 1 | 9.24 | 7 |
| Midway through the queue (30% of green) | 3 | 0 | 2 | 1 | 7.76 | |
| Towards queue end | 0 | 1 | 2 | 0 | 3.92 | |

MCLF traffic conditions, due to the smaller vehicles coming to the front of the queue and discharging first.

To assess the departure distribution at the two study sites, discharge data during the queue clearance times, at 3-second intervals were collected. Vehicular counts were converted into equivalent values using the PCE values and TCU values and tested for the distribution followed. The test statistics obtained for these two sites are given in Table 6, showing the probability of following the given distribution ($p$) and associated square error value.

It can be seen that, at Mhalgi nagar intersection, the discharge data followed uniform distribution, as indicated by higher $p$ value. On the other hand, the departure distribution at the Kaveli intersection was random aligning with Poisson distribution. The degree of saturation ($X$) was 0.85 at the Kaiveli intersection and 0.5 at the Mhalgi nagar intersection. To further assess the influence of varying $X$ values on the departure distribution, various trials were conducted taking Mhalgi nagar VISSIM network as sample, with $X$ values ranging from 0.6 to 0.9. The analysis revealed that for $X$ values up to 0.7, the departure distribution followed a uniform pattern. However, for $X$ values higher than 0.7, the distribution became random, adhering more closely to a Poisson distribution. This analysis summarizes that the departure distribution in MCLF traffic conditions is also stochastic at higher $X$ values.

**Theoretical delay model formulation.** The arrival and departure distribution revealed that, in MCLF traffic condition, both the arrivals and departures are random, particularly under peak conditions. In terms of number of servers multiple servers was found to be best choice. Based on these, using Kendall [21] notation, the best queueing model to represent MCLF traffic condition is M/M/n. Hence, delay or waiting time expressions were derived for these conditions, as detailed below.

The fundamental structure of any theoretical delay model includes a uniform delay term, random delay term, overflow delay term, and adjustment factors that are typically site specific. The current formulation of the delay model also follows the same structure. However, this study assumes that the intersection is under-saturated, and hence the overflow delay term, which accounts for delays caused by flow exceeding the capacity, is omitted.

The first term, known as the uniform delay term, is derived from the cumulative arrival and departure curves. This term represents the deterministic component of delay, as illustrated in Fig 9. In this context, the area between the cumulative arrivals and departures corresponds to the total delay experienced by all vehicles seeking to traverse through the intersection in a single cycle. Equations (5) can then be derived to calculate the average uniform delay ($d_1$) with the assumption of uniform arrivals.

**Table 6. Test statistics of random and uniform distribution check.**

| Study site | Random Distribution | | Uniform Distribution | |
|---|---|---|---|---|
| | p value | Square error | p value | Square error |
| **Mhalgi nagar** | 0.04 | 0.023 | 0.08 | 0.03 |
| **Kaiveli** | 0.02 | 0.011 | <0.005 | 0.03 |

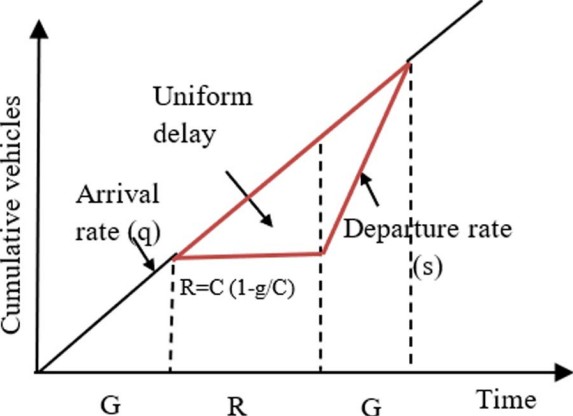

**Fig 9. Deterministic component of delay.**

$$d_1 = \frac{C\left(1-\dfrac{g}{C}\right)^2}{2\left(1-\dfrac{q}{s}\right)}. \tag{5}$$

Substituting $\lambda = g/C$ and $X = q/(sg/C)$, Equation (5) becomes:

$$d_1 = \frac{C(1-\lambda)^2}{2(1-\lambda X)}. \tag{6}$$

In MCLF traffic condition, when using the uniform delay term, it is essential to convert the arrival and departure vehicle count into equivalent values using PCE. This helps to accurately represent the impact of various vehicle types on traffic flow and delay, ensuring that the model reflects the true dynamics of multiple classes of vehicles.

Second term corresponds to the random delay, which accounts for the random fluctuations in the arrivals. In this condition, some cycles may fail, even though the whole intersection $q/c$ is less than one. Fig 10 illustrates the random delay, where dotted line represents the capacity. Area between the arrival curve and this dotted line depicts the random delay. In traditional models, this term is derived assuming random arrivals and deterministic departures. However, under MCLF conditions, the arrivals and departures follow random distribution. Incorporating all these, the delay expression was derived as discussed below.

As discussed earlier, a multi-channel queueing process is considered as the best way to represent the queue formation under MCLF traffic conditions. As the arrivals and departures are stochastic under MCLF traffic condition, the system is modelled as M/M/n system, which follows Poisson arrivals and departure with $n$ service channels. The corresponding random delay term ($d_2$) is derived based on this.

The basic derivation starts with the concept of birth-death process. One of the basic assumptions made in earlier studies [40] is the system is always operating at its full capacity. This assumption is not realistic for signalized intersection, as discharge occur only during green signals. Consequently, capacity is a more appropriate term for describing departure

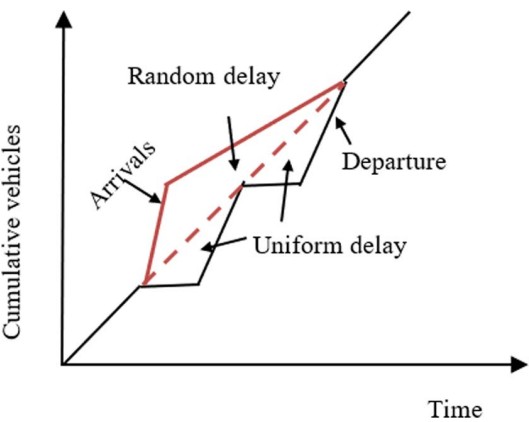

**Fig 10. Random component of delay.**

from a signalized intersection. Using this concept, for a multiple channel queueing system, there are four conditions at time *t* which must be considered in the formulation of transient equations, which are:

1. The system is empty with no entities ($N=0$).

2. The number of entities is less than the number parallel servers ($N<n$).

3. The system with number of entities and number of parallel servers are same ($N=n$).

4. The number of entities is higher than the number of parallel servers ($N>n$).

Let $P_0(t+\Delta t)$ and $P_N(t+\Delta t)$ be the probability that the queueing system has zero and $N$ entities at the time $+\Delta t$. There are three ways in which the system could have reached this state, assuming $\Delta t$ is so small that only one unit could have entered or departed.

a) The system did not change from time t to $t+\Delta t$

b) The system was in state *N-1* at *t*, and one arrived in $\Delta t$

c) The system was in state $N+1$ at *t*, and one departed in $\Delta t$

Since the probability of one arrival in $\Delta t$ equals $q\Delta t$ and the probability of one departure in $\Delta t$ is $c\Delta t$, the corresponding probabilities of zero arrivals and zero departures are $q\Delta t$ and $c\Delta t$. If higher orders of $\Delta t$ are neglected, the mathematical description of the three ways for the system to reach *N* and zero entities in $t+\Delta t$ of the above-stated four conditions of multiple channel queueing system are:

$$P_0(t+\Delta t)=P_0(t)(1-q\Delta t)+P_1(t)(1-q\Delta t)c\Delta t. \tag{7}$$

$$P_N(t+\Delta t)=P_N(t)(1-q\Delta t)(1-Nc\Delta t)+P_{N-1}(t)q\Delta t(1-(N-1)c\Delta t)$$
$$+P_{N+1}(t)(1-q\Delta t)(N+1)c\Delta t. \tag{8}$$

$$P_N(t+\Delta t)=P_N(t)(1-\lambda\Delta t)(1-nc\Delta t)+P_{N-1}(t)q\Delta t(1-(n-1)c\Delta t)$$
$$+P_{N+1}(t)(1-q\Delta t)nc\Delta t. \tag{9}$$

$$P_N(t+\Delta t) = P_N(t)(1-q\Delta t)(1-nc\Delta t) + P_{N-1}(t)q\Delta t(1-nc\Delta t) + P_{N+1}(t)(1-q\Delta t)nc\Delta t. \quad (10)$$

Equations (9) and (10) become the same when higher orders of $\Delta t$ are neglected, and so the transient equations become:

$$P_0'(t) = -qP_0(t) + cP_1(t). \quad (11)$$

$$P_N'(t) = -(q+Nc)P_N(t) + qP_{N-1}(t) + (N+1)cP_{N+1}(t). \quad (12)$$

$$P_N'(t) = -(q+nc)P_N(t) + qP_{N-1}(t) + ncP_{N+1}(t). \quad (13)$$

The steady state equations are obtained when the time derivatives are set to zero and equations are obtained as above only when $X = q/nc < 1$. The solution of these steady state equations by induction are:

$$P_N = \frac{(nX)^N}{N!} P_0 \quad \text{for } N < n \quad (14)$$

$$\text{and } P_N = \frac{(nX)^N}{n! \ n^{N-n}} P_0 \quad \text{for } N \geq n \quad (15)$$

$P_0$ can be found from,

$$\sum_{\infty}^{N=0} P_N = \sum_{n-1}^{N=0} P_N + \sum_{\infty}^{N=n} P_N = 1. \quad (16)$$

Solving for $P_0$ gives,

$$P_0 = \frac{1}{\sum_{N=0}^{n-1}\left[\frac{(nX)^N}{N!}\right] + \frac{(nX)^n}{n!(1-X)}}. \quad (17)$$

From these expected number in the queue can be calculated as,

$$E(m) = \sum_{\infty}^{N=n} (N-n)P_N = \sum_{\infty}^{N=n} (N-n)P_0 \frac{(nX)^N}{n! \ n^{N-n}}. \quad (18)$$

On simplification Equations (18) becomes,

$$E(m) = \frac{X}{n!} \frac{(nX)^n}{(1-X)^2} P_0. \quad (19)$$

From this, using the Little's queueing formula [53], the average waiting time in the queue is,

$$W_q = \frac{E(m)}{q} = \frac{X}{q \ n!} \frac{(nX)^n}{(1-X)^2} \frac{1}{\sum_{N=0}^{n-1}\left[\frac{(nX)^N}{N!}\right] + \frac{(nX)^n}{n!(1-X)}}. \quad (20)$$

Equation (20) represents the $d_2$ of the delay formulation. Then, the average control delay is obtained by the addition of uniform and random delay, and it is expressed as:

$$d = \frac{C(1-\lambda)^2}{2(1-\lambda X)} + \frac{X}{q}\frac{(nX)^n}{n!\,(1-X)^2}\frac{1}{\sum_{N=0}^{n-1}\left(\frac{(nX)^N}{N!}\right) + \frac{(nX)^n}{n!(1-X)}}. \tag{21}$$

Equation (21) represents the delay expression for MCLF under-saturated traffic condition. At the study sites, Mhalgi nagar and Kaiveli intersection, the number of parallel serves was 5 and 7 respectively. By substituting the respective values of $n$ into Equation (21), the delay equation for Mhalgi nagar and Kaiveli site were obtained and are given in (22) and (23) respectively.

$$d = \frac{C(1-\lambda)^2}{2(1-\lambda X)} + \frac{625X^6}{q(1-X)\left(24 + 96X + 180X^2 + 200X^3 + 125X^4\right)}. \tag{22}$$

$$d = \frac{C(1-\lambda)^2}{2(1-\lambda X)} + \frac{117649X^8}{q(1-X)\left(\begin{array}{l}720 + 4320X + 12600X^2 + 23520X^3 \\ + 30870X^4 + 28812X^5 + 16807X^6\end{array}\right)}. \tag{23}$$

It is essential to validate this model to ensure its accuracy and applicability in the real-world traffic scenario, and is explained in the following section.

## Model validation and results

The developed model was validated using the delay values calculated using the field data of Mhalgi nagar square intersection, Nagpur and Kaiveli Intersection, Chennai. Additionally, the performance of the model is evaluated using the various scenarios generated in the calibrated and validated VISSIM network of Mhalgi nagar square intersection.

### Model validation using field data

The delay values of each cycle obtained from the field data were compared with the delay estimates from the proposed delay model. The comparative assessment of average approach delays for Mhalgi nagar and Kaiveli are shown in Figs 11 and 12, respectively.

From these figures, it is evident that the proposed delay model effectively captures the actual delay values under both field conditions. The average Mean Absolute Percentage Error (MAPE) for the delay values at Mhalgi nagar, Nagpur, was 14%. Among the 10 observed cycles at Kaiveli, 2 cycles were over-saturated, where the applicability of the steady-state stochastic model is reported to be limited. Hence, the comparative results for the remaining 8 cycles are presented in Fig 12, with each cycle exhibiting good accuracy between the field-observed and model-predicted values. The average MAPE for these cycles was 12%.

### Model validation using simulated data and sensitivity analysis

To better understand the performance of the proposed delay model at other $X$ values, scenarios with $X$ values ranging from 0.5 to 0.9 were generated in the Mhalgi nagar VISSIM network. The delay results obtained from the proposed delay model were compared against those obtained from VISSIM, Webster's delay model [22], Akcelik delay model [31], HCM model [35] and Indo-HCM delay model [43]. The MAPE values of each delay model compared to the VISSIM delay were calculated and illustrated in Fig 13.

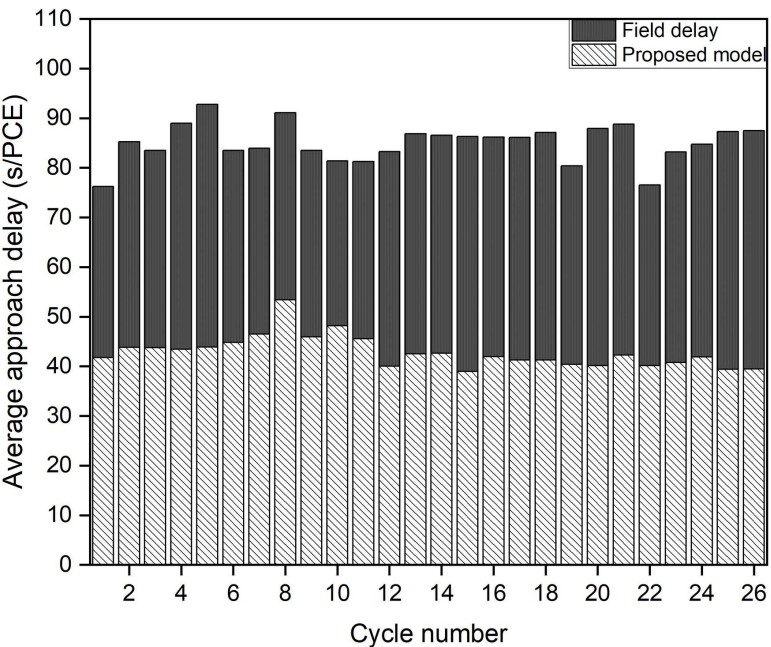

**Fig 11. Cycle by cycle comparison of delay estimates of Proposed model with field data of Mhalgi nagar, Nagpur site.**

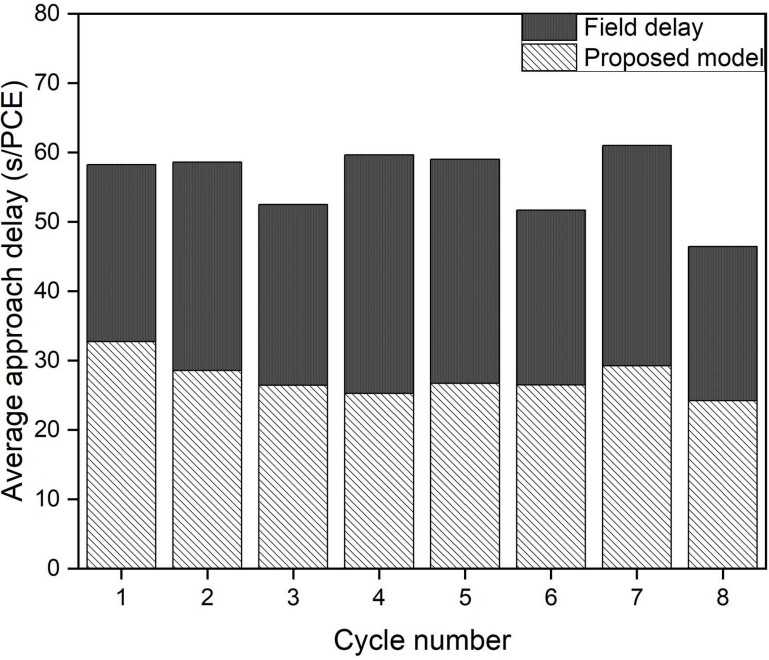

**Fig 12. Cycle by cycle comparison of delay estimates of Proposed model with field data of Kaiveli, Chennai site.**

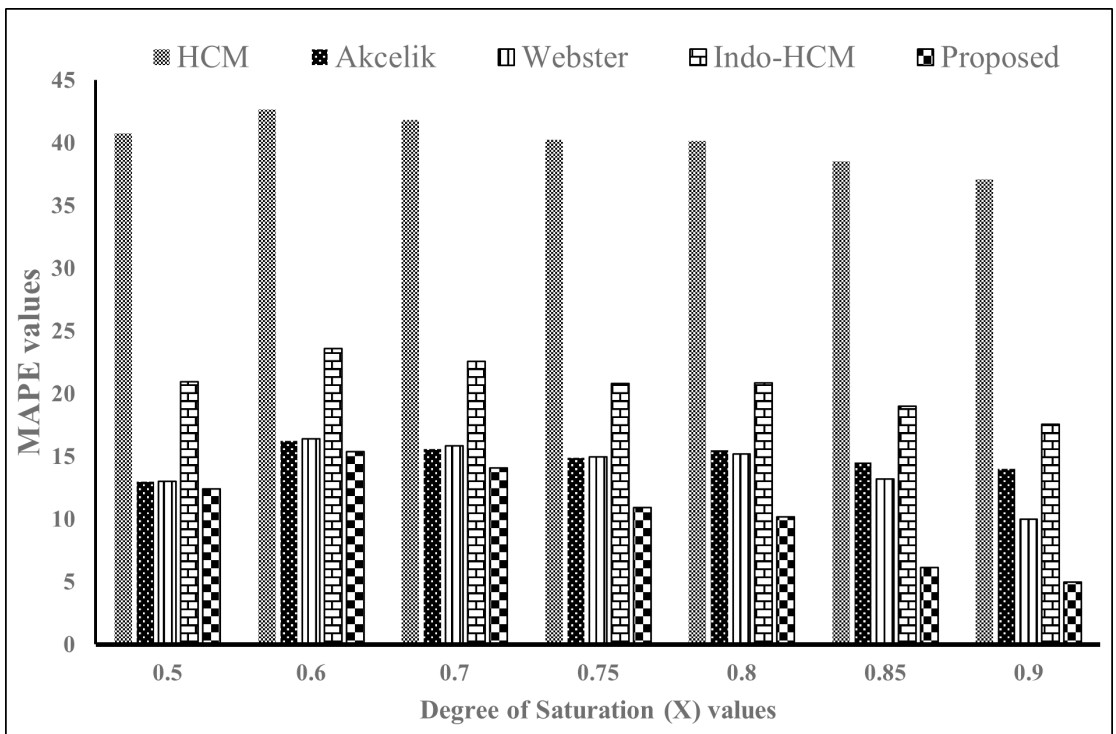

**Fig 13. MAPE values of delay models compared with VISSIM delay.**

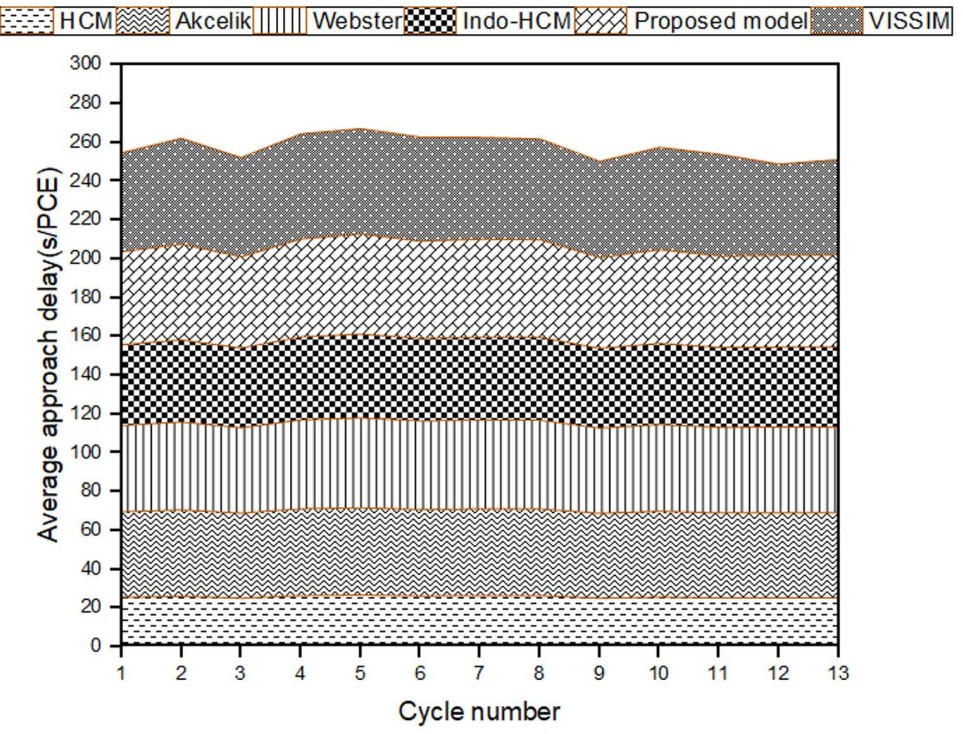

**Fig 14. Delay estimates of analytical models and VISSIM at X value of 0.9.**

In all scenarios considered, the proposed delay model demonstrated superior performance as evidenced by lower MAPE values. The MAPE value of the proposed delay model decreased from 0.5 to 0.9, indicating its coherence to actual delay at higher *X* values. The most significant result was observed at an *X* value of 0.9. The comparison of results at an *X* value of 0.9 is depicted in Fig 14. It can be seen that the delay estimated using the proposed delay model is the closest to the actual delay obtained from VISSIM. The MAPE values associated with the delay estimate were 13%, 15%, 50%, 15% and 6% for Webster, Akcelik, HCM, Indo-HCM and proposed delay model respectively. The corresponding Mean Absolute Error (MAE) values were 6.90, 7.59, 25.96, 9.92 and 3.16 respectively. Fig 15 shows the MAPE and MAE values for each model. Additionally, Fig 16 illustrates the actual and estimated values of delay, reflecting the better coherence of proposed delay model in estimating delay under MCLF traffic condition. These results signify the superior performance of the proposed delay model in MCLF traffic condition, demonstrating its robustness and accuracy compared to existing conventional models.

## Conclusion

Delay is a critical metric for assessing signalized intersection performance and is commonly used as a parameter for optimizing signal design. However, accurate measurement of delay is difficult due to the random nature of traffic arrivals and departures. Analytical delay models that have been proposed, primarily tailored for HoLD traffic conditions, do not incorporate characteristics of multi-class vehicles and a lack of lane discipline. Hence, there is a need for a delay model specifically tailored to such traffic conditions existing in several countries including India.

This study developed a theoretical delay model using the concepts of queueing system considering factors such as arrival and departure rate, patterns, number of servers, distribution of arrivals and departures as well as queueing models tailored to the MCLF traffic conditions. It integrates the concept of PCE values, virtual lanes, random arrival and departures, and multi-channel queueing system to address the complexities of MCLF traffic. The uniform

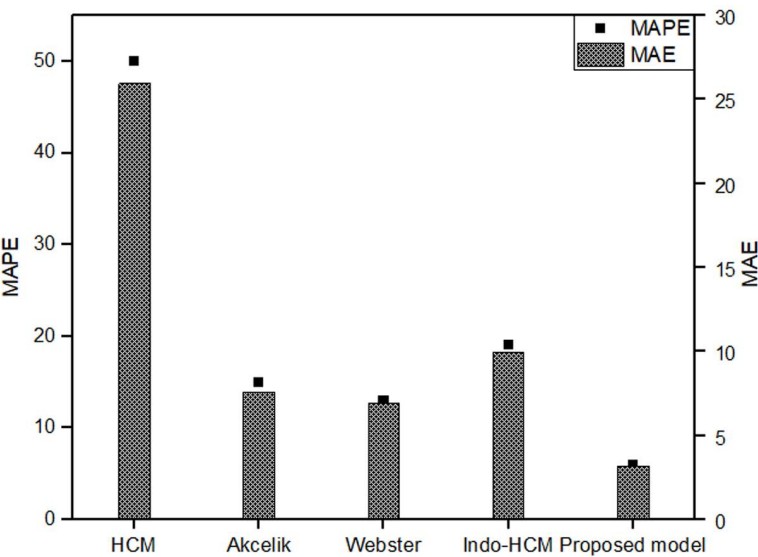

**Fig 15. MAPE and MAE values of delay models.**

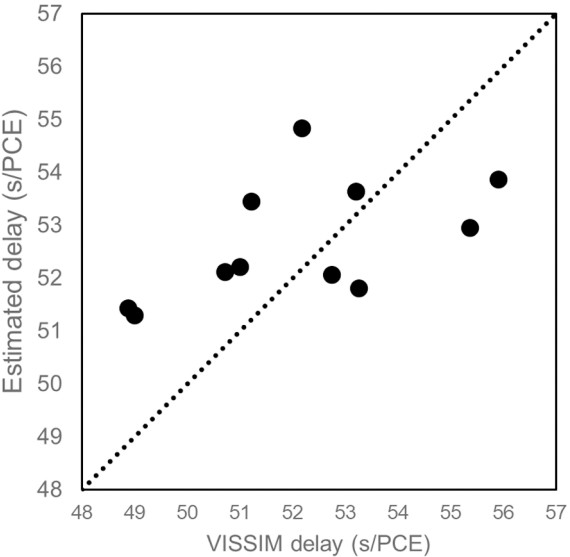

**Fig 16. Comparison of actual delay with estimated delay.**

delay term is derived from cumulative count curves, while the second term, representing random delay, is derived based on birth-death process of queueing analysis.

The developed delay model was validated using data from two study sites, for varying conditions simulated in VISSIM. Performance comparison showed a 44% reduction in the MAPE of delay estimates compared to conventional delay models in practice. It indicates the necessity of delay models explicitly tailored for MCLF traffic conditions. As the signal cycle length is primarily derived with the objective of minimising delay, future research can focus on optimising cycle length using the proposed delay model in MCLF traffic conditions.

## Supporting information

**S1 File. Collected data from the field and data used in the simulation.** (XLSX)

## Acknowledgment

Authors acknowledge CSIR-Central Road Research Institute (CRRI), New Delhi, India for providing the traffic video for the study site 1. The authors acknowledge the Chennai traffic police for granting permission for video data collection for study site 2.

## Author contributions

**Conceptualization:** Vinaya S. Mattungal, Lelitha Devi Vanajakshi.

**Data curation:** Vinaya S. Mattungal.

**Formal analysis:** Vinaya S. Mattungal, Lelitha Devi Vanajakshi.

**Funding acquisition:** Lelitha Devi Vanajakshi.

**Investigation:** Vinaya S. Mattungal, Lelitha Devi Vanajakshi.

**Methodology:** Vinaya S. Mattungal, Lelitha Devi Vanajakshi.

**Project administration:** Lelitha Devi Vanajakshi.

**Resources:** Lelitha Devi Vanajakshi.

**Software:** Lelitha Devi Vanajakshi.

**Supervision:** Lelitha Devi Vanajakshi.

**Validation:** Vinaya S. Mattungal, Lelitha Devi Vanajakshi.

**Visualization:** Vinaya S. Mattungal.

**Writing – original draft:** Vinaya S Mattungal.

**Writing – review & editing:** Lelitha Devi Vanajakshi.

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
