## [Decision Letter · Decision Letter 0]

10 Sep 2024

PONE-D-24-24169An Analytical Delay model for Multi class and Lane free Traffic conditionPLOS ONE

Dear Dr. Vanajakshi,

Thank you for submitting your manuscript to PLOS ONE. After careful consideration, we feel that it has merit but does not fully meet PLOS ONE’s publication criteria as it currently stands. Therefore, we invite you to submit a revised version of the manuscript that addresses the points raised during the review process. Please submit your revised manuscript by Oct 25 2024 11:59PM. If you will need more time than this to complete your revisions, please reply to this message or contact the journal office at plosone@plos.org . Please include the following items when submitting your revised manuscript:

We look forward to receiving your revised manuscript.

Kind regards,

Zhixia Li, Ph.D.

Academic Editor

PLOS ONE

Journal Requirements: When submitting your revision, we need you to address these additional requirements. 1. Please ensure that your manuscript meets PLOS ONE's style requirements, including those for file naming. The PLOS ONE style templates can be found at https://journals.plos.org/plosone/s/file?id=wjVg/PLOSOne_formatting_sample_main_body.pdf and https://journals.plos.org/plosone/s/file?id=ba62/PLOSOne_formatting_sample_title_authors_affiliations.pdf 2. In your Methods section, please provide additional information regarding the permits you obtained for the work. Please ensure you have included the full name of the authority that approved the field site access and, if no permits were required, a brief statement explaining why. 3. Thank you for stating the following financial disclosure: "Ministry of Road Transport and Highways (MoRTH) " Please state what role the funders took in the study.  If the funders had no role, please state: ""The funders had no role in study design, data collection and analysis, decision to publish, or preparation of the manuscript."" If this statement is not correct you must amend it as needed. Please include this amended Role of Funder statement in your cover letter; we will change the online submission form on your behalf. 4. In the online submission form, you indicated that "Available upon request" All PLOS journals now require all data underlying the findings described in their manuscript to be freely available to other researchers, either 1. In a public repository, 2. Within the manuscript itself, or 3. Uploaded as supplementary information.This policy applies to all data except where public deposition would breach compliance with the protocol approved by your research ethics board. If your data cannot be made publicly available for ethical or legal reasons (e.g., public availability would compromise patient privacy), please explain your reasons on resubmission and your exemption request will be escalated for approval. 5. We note that Figures 1(a) and 2(a) in your submission contain [map/satellite] images which may be copyrighted. All PLOS content is published under the Creative Commons Attribution License (CC BY 4.0), which means that the manuscript, images, and Supporting Information files will be freely available online, and any third party is permitted to access, download, copy, distribute, and use these materials in any way, even commercially, with proper attribution. For these reasons, we cannot publish previously copyrighted maps or satellite images created using proprietary data, such as Google software (Google Maps, Street View, and Earth). For more information, see our copyright guidelines: http://journals.plos.org/plosone/s/licenses-and-copyright. We require you to either (1) present written permission from the copyright holder to publish these figures specifically under the CC BY 4.0 license, or (2) remove the figures from your submission: A. You may seek permission from the original copyright holder of Figures 1(a) and 2(a) to publish the content specifically under the CC BY 4.0 license.   We recommend that you contact the original copyright holder with the Content Permission Form (http://journals.plos.org/plosone/s/file?id=7c09/content-permission-form.pdf) and the following text:“I request permission for the open-access journal PLOS ONE to publish XXX under the Creative Commons Attribution License (CCAL) CC BY 4.0 (http://creativecommons.org/licenses/by/4.0/). Please be aware that this license allows unrestricted use and distribution, even commercially, by third parties. Please reply and provide explicit written permission to publish XXX under a CC BY license and complete the attached form.” Please upload the completed Content Permission Form or other proof of granted permissions as an ""Other"" file with your submission. In the figure caption of the copyrighted figure, please include the following text: “Reprinted from [ref] under a CC BY license, with permission from [name of publisher], original copyright [original copyright year].” B. If you are unable to obtain permission from the original copyright holder to publish these figures under the CC BY 4.0 license or if the copyright holder’s requirements are incompatible with the CC BY 4.0 license, please either i) remove the figure or ii) supply a replacement figure that complies with the CC BY 4.0 license. Please check copyright information on all replacement figures and update the figure caption with source information. If applicable, please specify in the figure caption text when a figure is similar but not identical to the original image and is therefore for illustrative purposes only.The following resources for replacing copyrighted map figures may be helpful: USGS National Map Viewer (public domain): http://viewer.nationalmap.gov/viewer/ The Gateway to Astronaut Photography of Earth (public domain): http://eol.jsc.nasa.gov/sseop/clickmap/Maps at the CIA (public domain): https://www.cia.gov/library/publications/the-world-factbook/index.html and https://www.cia.gov/library/publications/cia-maps-publications/index.html NASA Earth Observatory (public domain): http://earthobservatory.nasa.gov/Landsat: http://landsat.visibleearth.nasa.gov/USGS EROS (Earth Resources Observatory and Science (EROS) Center) (public domain): http://eros.usgs.gov/#Natural Earth (public domain): http://www.naturalearthdata.com/

Reviewers' comments:

Reviewer's Responses to Questions

**Comments to the Author**

1. Is the manuscript technically sound, and do the data support the conclusions?

Reviewer #1: Partly

Reviewer #2: Yes

2. Has the statistical analysis been performed appropriately and rigorously? 

Reviewer #1: Yes

Reviewer #2: Yes

3. Have the authors made all data underlying the findings in their manuscript fully available?

Reviewer #1: No

Reviewer #2: No

4. Is the manuscript presented in an intelligible fashion and written in standard English?

Reviewer #1: Yes

Reviewer #2: Yes

5. Review Comments to the Author

Reviewer #1: This paper aims to propose a delay model for multi class and lane free traffic condition. Overall, I agree that a different delay model is needed under the new traffic operation mode, i.e., lane free traffic. My detailed comments are as follows:

1. It is stated that “Key features include ... lane-free movement prevalent in Indian traffic.” Please clarify how to consider the lane-free movement?

2. Some recent studies have described the two-dimensional movement. E.g., Developing an optimal intersection control system for automated connected vehicles, IEEE T-ITS, 2018; Unprotected left-turn behavior model capturing path variations at intersections, IEEE T-ITS, 2023; A behaviourally underpinned approach for two-dimensional vehicular trajectory reconstruction with constrained optimal control, TR-Part C, 2024. However, the authors did not mention them at all.

3. Model validation was not conducted using actual data, but rather simulation data was used as ground truth. Why can simulation be considered as ground truth?

4. Are the virtual lane numbers dynamically changing? According to the author's assumption, the virtual lane numbers seem to vary under different circumstances. How were they handled during model development?

5. How were the parameters in the theoretical model calibrated?

6. How are the datasets for model calibration and model testing distinguished?

Reviewer #2: 1. Literature Review is very short. A significant number of papers discussing lane-free roads should be included in this section.

please add and provide an explanation of the articles listed below:

• Papageorgiou, M., Mountakis, K.S., Karafyllis, I., Papamichail, I., Wang, Y.: Lane-free artificial-fluid concept for vehicular traffic, Proceedings of the IEEE 109 (2021), pp. 114-121. -

• Malekzadeh, M., Papamichail, I., Papageorgiou, M., Bogenberger, K.: Optimal internal boundary control of lane-free automated vehicle traffic, Transportation Research Part C 126 (2021), Article 103060.

• Malekzadeh, M., Papamichail, I., Papageorgiou, M.: Linear-Quadratic regulators for internal boundary control of lane-free automated vehicle traffic, Control Engineering Practice 115 (2021), Article 104912.

• Karafyllis, I., Theodosis, D., Papageorgiou, M.: Analysis and control of a non-local PDE traffic flow model, International Journal of Control 95 (2022), pp. 660-678. -

• Karafyllis, I., Theodosis, D., Papageorgiou, M.: Lyapunov-based two-dimensional cruise control of autonomous vehicles on lane-free roads, Automatica 145 (2022), Article 110517.

• Karafyllis, I., Theodosis, D., Papageorgiou, M.: Constructing artificial traffic fluids by designing cruise controllers, Systems & Control Letters 167 (2022), Article 105317. -

• Karafyllis, I., Theodosis, D., Papageorgiou, M.: Nonlinear adaptive cruise control of vehicular platoons, International Journal of Control 96 (2023), pp. 147-169. - -

• Malekzadeh, M., Yanumula, V.K., Papamichail, I., Papageorgiou, M.: Overlapping internal boundary control of lane-free automated vehicle traffic, Control Engineering Practice 133 (2023), Article 105435.

• Yanumula, V.K., Typaldos, P., Troullinos, D., Malekzadeh, M., Papamichail, I., Papageorgiou, M.: Optimal trajectory planning for connected and automated vehicles in lane-free traffic with vehicle nudging, IEEE Transactions on Intelligent Vehicle 8 (2023), pp. 2385-2399. -

• Karafyllis, I., Papageorgiou, M.: A particle method for 1-D compressible fluid flow, Studies in Applied Mathematics 151 (2023), pp. 1282–1331.

• Malekzadeh, M., Troullinos, D., Papamichail, I., Papageorgiou, M., Bogenberger, K.: Internal boundary control in lane-free automated vehicle traffic: Comparison of approaches via microscopic simulation, Transportation Research Part C 158 (2024), Article 104456.

• Naderi, M., Papageorgiou, M., Troullinos, D., Karafyllis, I., Papamichail, I.: Controlling automated vehicles on large lane-free roundabouts, IEEE Transactions on Intelligent Vehicle 9 (2024), pp. 3061-3074.

• Jin, X., Yu, X., Hu, Y., Wang, Y., Papageorgiou, M., Papamichail, I., Malekzadeh, M.: Integrated control of internal boundary and ramp inflows for lane-free traffic of automated vehicles on freeways. 25th IEEE International Conference on Intelligent Transportation Systems (ITSC 2022), Macau, China, October 8-12, 2022, pp. 1234-1239

• Malekzadeh, M., Manolis, D., Papamichail, I., Papageorgiou, M.: Empirical investigation of properties of lane-free automated vehicle traffic. 25th IEEE International Conference on Intelligent Transportation Systems (ITSC 2022), Macau, China, October 8-12, 2022, pp. 2393-2400.

• Papamichail, I., Schoenn-Anchling, N., Malekzadeh, M., Markantonakis, V., Papageorgiou, M.: Macroscopic traffic flow model calibration for lane-free automated vehicle traffic. 26th IEEE International Conference on Intelligent Transportation Systems (ITSC 2023), Bilbao, Bizkaia, Spain, September 24-28, 2023, pp. 3485-3492

2. The quality of the fihures are not good. Please improve it.

3. Plaese explain more about the equations.

6. PLOS authors have the option to publish the peer review history of their article (what does this mean? ). If published, this will include your full peer review and any attached files.

**Do you want your identity to be public for this peer review?** For information about this choice, including consent withdrawal, please see our Privacy Policy .

Reviewer #1: No

Reviewer #2: No

---

## [Decision Letter · Decision Letter 1]

28 Nov 2024

PONE-D-24-24169R1An Analytical Delay model for Multi class and Lane free Traffic conditionPLOS ONE

Dear Dr. Vanajakshi,

Thank you for submitting your manuscript to PLOS ONE. After careful consideration, we feel that it has merit but does not fully meet PLOS ONE’s publication criteria as it currently stands. Therefore, we invite you to submit a revised version of the manuscript that addresses the points raised during the review process.

We look forward to receiving your revised manuscript.

Kind regards,

Zhixia Li, Ph.D.

Academic Editor

PLOS ONE

Journal Requirements:

Additional Editor Comments:

The manuscript has been thoroughly revised. The authors would need to respond/address Reviewer 2's comments.

Reviewers' comments:

Reviewer's Responses to Questions

**Comments to the Author**

1. If the authors have adequately addressed your comments raised in a previous round of review and you feel that this manuscript is now acceptable for publication, you may indicate that here to bypass the “Comments to the Author” section, enter your conflict of interest statement in the “Confidential to Editor” section, and submit your "Accept" recommendation.

Reviewer #1: All comments have been addressed

Reviewer #2: (No Response)

2. Is the manuscript technically sound, and do the data support the conclusions?

Reviewer #1: Yes

Reviewer #2: Partly

3. Has the statistical analysis been performed appropriately and rigorously? 

Reviewer #1: Yes

Reviewer #2: I Don't Know

4. Have the authors made all data underlying the findings in their manuscript fully available?

Reviewer #1: No

Reviewer #2: Yes

5. Is the manuscript presented in an intelligible fashion and written in standard English?

Reviewer #1: Yes

Reviewer #2: Yes

6. Review Comments to the Author

Reviewer #1: I suggestions have been properly addressed. The current version can be published. I have no more comments.

Reviewer #2: The authors did not address my comments properly. I requested that they enhance the literature review and clarify the equation explanations, but they did not comply.

7. PLOS authors have the option to publish the peer review history of their article (what does this mean? ). If published, this will include your full peer review and any attached files.

**Do you want your identity to be public for this peer review?** For information about this choice, including consent withdrawal, please see our Privacy Policy .

Reviewer #1: No

Reviewer #2: No

---

## [Decision Letter · Decision Letter 2]

31 Jan 2025

An Analytical Delay model for Multi class and Lane free Traffic condition

PONE-D-24-24169R2

Dear Dr. Vanajakshi,

We’re pleased to inform you that your manuscript has been judged scientifically suitable for publication and will be formally accepted for publication once it meets all outstanding technical requirements.

Kind regards,

Zhixia Li, Ph.D.

Academic Editor

PLOS ONE

Additional Editor Comments (optional):

Reviewers' comments:

Reviewer's Responses to Questions

**Comments to the Author**

1. If the authors have adequately addressed your comments raised in a previous round of review and you feel that this manuscript is now acceptable for publication, you may indicate that here to bypass the “Comments to the Author” section, enter your conflict of interest statement in the “Confidential to Editor” section, and submit your "Accept" recommendation.

Reviewer #1: All comments have been addressed

Reviewer #3: All comments have been addressed

2. Is the manuscript technically sound, and do the data support the conclusions?

Reviewer #1: Yes

Reviewer #3: Yes

3. Has the statistical analysis been performed appropriately and rigorously? 

Reviewer #1: Yes

Reviewer #3: N/A

4. Have the authors made all data underlying the findings in their manuscript fully available?

Reviewer #1: Yes

Reviewer #3: Yes

5. Is the manuscript presented in an intelligible fashion and written in standard English?

Reviewer #1: Yes

Reviewer #3: Yes

6. Review Comments to the Author

Reviewer #1: My comments have been adequately addressed. I have no more comments. The manuscript can be pubished in this version.

Reviewer #3: (No Response)

7. PLOS authors have the option to publish the peer review history of their article (what does this mean? ). If published, this will include your full peer review and any attached files.

**Do you want your identity to be public for this peer review?** For information about this choice, including consent withdrawal, please see our Privacy Policy .

Reviewer #1: No

Reviewer #3: No
